# Radiotherapy in the Management of Gastrointestinal Stromal Tumors: A Systematic Review

**DOI:** 10.3390/cancers14133169

**Published:** 2022-06-28

**Authors:** Haidong Zhang, Tianxiang Jiang, Mingchun Mu, Zhou Zhao, Xiaonan Yin, Zhaolun Cai, Bo Zhang, Yuan Yin

**Affiliations:** 1Department of Gastrointestinal Surgery, West China Hospital, Sichuan University, Chengdu 610041, China; zhd3287332910@163.com (H.Z.); jiangtx98@163.com (T.J.); mmcmmcmmc163@163.com (M.M.); zhaozhou_med@163.com (Z.Z.); yxnyinxiaonan@163.com (X.Y.); caizhaolun@foxmail.com (Z.C.); 2Department of Gastrointestinal Surgery, Sanya People’s Hospital, West China Sanya Hospital, Sichuan University, Sanya 572000, China

**Keywords:** gastrointestinal stromal tumor, GIST, management, radiotherapy, radiation therapy, symptom palliation, adverse events

## Abstract

**Simple Summary:**

Gastrointestinal stromal tumors are considered to be insensitive to radiotherapy. However, with the development of radiation techniques and the accumulation of cases, some studies have indicated that radiotherapy could help achieve objective response in advanced or metastatic gastrointestinal stromal tumors. Therefore, it is necessary to conduct a systematic review to reassess the role of radiotherapy in gastrointestinal stromal tumors. The purpose of this study was to draw the attention of scholars and clinicians to radiotherapy and promote further research on radiotherapy in gastrointestinal stromal tumors.

**Abstract:**

Gastrointestinal stromal tumors (GISTs) are considered insensitive to radiotherapy. However, a growing number of case reports and case series have shown that some lesions treated by radiotherapy achieved an objective response. The aim of the study was to perform a systematic review of all reported cases, case series, and clinical studies of GISTs treated with radiotherapy to reevaluate the role of radiotherapy in GISTs. A systematic search of the English-written literature was conducted using PubMed, Web of Science, and Embase databases. Overall, 41 articles describing 112 patients were retrieved. The included articles were of low to moderate quality. Bone was the most common site treated by radiotherapy, followed by the abdomen. In order to exclude the influence of effective tyrosine kinase inhibitors (TKIs), a subgroup analysis was conducted on whether and which TKIs were concurrently applied with radiotherapy. Results showed that radiotherapy alone or combined with resistant TKIs could help achieve objective response in selected patients with advanced or metastatic GISTs; however, survival benefits were not observed in the included studies. Pain was the most common symptom in symptomatic GISTs, followed by neurological dysfunction and bleeding. The symptom palliation rate was 78.6% after excluding the influence of effective TKIs. The adverse reactions were mainly graded 1–2. Radiotherapy was generally well-tolerated. Overall, radiotherapy may relieve symptoms for GIST patients with advanced or metastatic lesions and even help achieve objective response in selected patients without significantly reducing the quality of life. In addition to bone metastases, fixed abdominal lesions may be treated by radiotherapy. Publication bias and insufficient quality of included studies were the main limitations in this review. Further clinical studies are needed and justified.

## 1. Introduction

Gastrointestinal stromal tumor (GIST) is the most common mesenchymal tumor in the gastrointestinal tract [1], with significant variations in reported incidence (from 0.4 to 2 cases per 100,000 per year [2,3]. The most common site of GISTs is the stomach, followed by the small intestine, which is now thought to originate from interstitial cells of Cajal (ICC) [4]. Functional mutations in the *KIT* gene and *PDGFRA* gene drive approximately 90% of GISTs [5]. At present, complete surgical resection is the standard treatment for locoregional lesions. Adjuvant 3-year imatinib therapy is given after surgery for GISTs, with significant recurrence risk. In contrast, the standard treatment for advanced, inoperable, and metastatic disease is tyrosine kinase inhibitors (TKIs) [2]. Although the use of molecularly targeted drugs such as imatinib significantly prolonged the overall survival of patients with GISTs [6,7], local treatment, such as surgery, radiofrequency ablation, and hepatic artery embolization, may be recommended for selected patients with advanced or metastatic GISTs [8]. In the past, GISTs were considered insensitive to radiotherapy [9], which is recommended for palliative intent in patients with advanced lesions or metastatic disease [2]. However, with the development of radiotherapy technology, some published cases and case series have shown that radiotherapy may be used for therapeutic purposes [10,11]. Radiotherapy is rarely used in GISTs, and the literature is limited to case reports and a few clinical studies with a limited number of cases. Therefore, a systematic review of the literature synthesizing these reports helps physicians by providing the best evidence for reassessing radiotherapy in the management of GISTs. The aim of the present study was to perform a systematic review of all reported radiation-treated cases.

## 2. Methods and Data Management

### 2.1. Protocol

This systematic review was reported according to the Preferred Reporting Items for Systematic Reviews and Meta-Analyses (PRISMA) checklist [12] (Appendix A). This protocol was prospectively registered in the Open Science Framework Registry (https://osf.io/qba6j, accessed on 22 June 2022).

### 2.2. Study Design

A systematic review was performed that analyzed radiotherapy in the management of GISTs to answer the following question: “What is the potential value of radiotherapy in GISTs”?

### 2.3. Eligibility Criteria

#### 2.3.1. Inclusion Criteria

Articles in which patients with confirmed GISTs were treated with radiotherapy combined with/without TKIs and/or surgery were included, irrespective of what type of treatment the patients previously had. Case series were defined as reports on treatment outcomes in more than 2 patients. In addition, at least one of the following was obtained from the included articles: (1) patient response to radiotherapy; (2) duration of disease control (time to progression, time to recurrence, and survival); (3) symptom palliation; (4) adverse events.

#### 2.3.2. Exclusion Criteria

Studies written in non-English languages, cases with synchronous or heterochronous tumors, case series with other types of tumors and reviews, and unavailable full texts were excluded.

### 2.4. Information Sources and Search Strategy

A systematic review of the English-written literature was conducted using PubMed, Web of Science, and Embase databases, with individual search strategies for each database. A comprehensive search was undertaken to retrieve original studies using the keywords gastrointestinal stromal tumor, GIST, radiotherapy, and their variations. No time limit was imposed on publication dates. The last search was performed on 18 May 2022. The reference lists of all relevant articles were scanned to identify any possible related studies to be included [13].

### 2.5. Study Selection

The selection was completed in two phases. In phase one, all retrieved abstracts were screened by two authors (H.Z. and T.J.). For each one that met the inclusion criteria, the full text was obtained. In phase two, full-text reading was performed independently by the same two authors. They had discussions to reach a consensus when disagreements arose. When a consensus was not reached, a third author (Y.Y.) was involved in making a final decision.

### 2.6. Methodological Quality Assessment of Included Studies

The Joanna Briggs Institute (JBI) Critical Appraisal Checklist for Case Reports and Case series and the CARE Checklist were adapted and applied for the methodological quality assessment [14,15]. Regarding JBI quality appraisal, two reviewers (H.Z. and T.J.) scored 9 items, including whether to report according to the CARE Checklist, as “yes”, “no”, “unclear”, and “not applicable” for case reports, and 10 items for case series. Any disagreement was resolved by consensus or the decision of a third author (Y.Y.). The quality evaluation results were divided into three grades: low, moderate, and high. In case reports, we attached more importance to the details of diagnosis, treatment procedures, and effects. Therefore, “low quality” was defined as not all of items 4, 5, and 6 receiving a “yes” response. “Moderate quality” was defined as all items 4, 5, and 6 receiving “yes” while not all other nine items receiving a “yes” score. “High quality” was defined as all nine items receiving “yes”. For item 1, we gave “yes” to case reports in which the age and sex could at least be obtained. In addition, the histological results—namely, immunohistochemical analysis for item 4; radiation dose and fractions for item 5; and symptom palliation, response to radiotherapy, or recurrence for item 6—could also at least be retrieved. In addition, in case series, we attached more importance to inclusion criteria, diagnosis, consistent inclusion, treatment procedure, and outcomes or follow-up. Therefore, “low quality” was defined as not all items 1, 3, 4, 7, and 8 receiving a “yes” response. “Moderate quality” was defined as all items 1, 3, 4, 7, and 8 receiving a “yes” score while not 10 items receiving “yes”. “High quality” was defined as all 10 items receiving a “yes” response (Appendix A).

### 2.7. Data Collection Process and Data Items

Age, sex, sites treated by radiotherapy, dose and fractions, previous and concomitant TKIs, symptom palliation, adverse events, disease response, time to progression, time to recurrence, and survival time were recorded by one author (H.Z.). A second author (T.J.) cross-checked all the collected information. Again, any disagreement was resolved by consensus or the decision of a third author (Y.Y.). The results of response to radiotherapy and recurrence should be supported by objective images (pre- and post-treatment images) or based on definite criteria that were presented in articles. Regarding response, we defined articles in which definite criteria and objective images were not presented as “not available”. When the authors evaluated a response according to specific criteria or images presented in studies, we accepted it. When the articles presented images and did not evaluate responses to radiotherapy, we evaluated the responses based on the images according to Response Evaluation Criteria in Solid Tumors (RECIST criteria).

### 2.8. Outcomes of Interest

The included studies were synthesized in qualitative and quantitative descriptions. Response and symptom palliation after radiotherapy were the primary outcomes. In addition, we defined the initiation of radiotherapy as the starting point for follow-up. Overall survival (OS) was calculated from the date of radiotherapy initiation to the date of death. Time to progression and recurrence were calculated from radiotherapy initiation to progression and recurrence, respectively. Local progression and recurrence were defined as any clinical or radiographic evidence of tumor growth. Local progression-free survival (PFS), local recurrence-free survival (RFS), and OS were estimated using the Kaplan–Meier method. OS, PFS, RFS, and adverse events related to radiotherapy were the secondary outcomes.

## 3. Results

### 3.1. Study Selection

Finally, 412 studies were retrieved from the 3 electronic databases, and 2 were obtained from reference lists. Then, duplicate articles were removed, resulting in 315 remaining studies. Then, a comprehensive evaluation of the abstracts was conducted, and 265 articles were excluded. Therefore, 50 manuscripts were selected for full-text review. Later, four case reports were excluded due to reporting GISTs with synchronous or heterochronous tumors, and five case series were excluded due to reporting GISTs with other types of tumors. There were a total of 41 retrieved articles describing 112 patients [10,11,16,17,18,19,20,21,22,23,24,25,26,27,28,29,30,31,32,33,34,35,36,37,38,39,40,41,42,43,44,45,46,47,48,49,50,51,52,53,54] for qualitative analysis (Figure 1). Among them, 35 articles were case reports (Table 1), and 6 were case series (Table 2). According to the quality assessment, there were 20 low-quality and 15 moderate-quality case reports. There were five low-quality and one moderate-quality case series. There were no high-quality studies in either case reports or case series (Appendix A). These patients consisted of 36 females (32.1%) and 76 males (67.9%), with ages ranging from 19.7 to 86.5 years.

### 3.2. Patient Response to Radiation and Follow-Up

There were 34 case reports and 2 case series, covering 70 lesions in 55 patients, which clearly described the patients’ responses to radiotherapy and the specific scenarios of radiotherapy combined with TKIs [10,11,16,17,18,19,20,21,22,23,24,25,26,27,28,29,30,31,32,33,34,35,36,37,38,39,40,41,42,43,44,45,46,50,51,52]. The total doses of radiation ranged from 15 Gy to 85 Gy. The most common pattern was 30 Gy in 10 fractions. We divided the 70 lesions into 2 parts: 53 defined irradiated lesions in 41 patients (specific lesions in images or macroscopic incompletely resected lesions) and 17 undefined lesions in 17 patients (macroscopic completely resected lesions; radiotherapy was used as adjuvant therapy after complete resection).

We divided the 53 defined irradiated lesions into 4 groups according to radiotherapy with/without concomitant TKIs: radiotherapy (R), radiotherapy with new TKIs (R + nT, radiotherapy with further lines of TKIs after resistance), radiotherapy with resistant TKIs (R + rT, radiotherapy with previous resistant TKIs), and radiotherapy with sensitive TKIs (R + sT, radiotherapy with imatinib in cases in which no TKIs have been used before). The responses of the lesions are presented in Table 3. There were a total of 32 evaluable lesions. In particular, in the “R” group, partial response (PR) was observed in six lesions, and stable disease (SD) was observed in six lesions. In addition, in the “R + rT” group, complete response (CR) was seen in one lesion, PR in four, and SD in five. We further analyzed the locations of these 53 lesions, and the results are presented in Table 4. Bone and joints (26/53) were the most common sites treated by radiotherapy, followed by the abdomen (14/53).

Among the 41 patients who had defined lesions treated by radiotherapy, Cuaron et al. reported 15 patients with locally advanced or metastatic GISTs [11]. There were 12 deaths, with a median follow-up of 5.1 months (range, 1.4–28.3). The estimated 6-month local progression-free survival was 57.0%. The median survival was 6.6 months, and the estimated 6-month overall survival was 57.8%. Among the remaining 26 patients who were from 24 case reports [10,17,18,19,20,21,22,23,26,28,29,30,31,32,34,35,37,39,41,43,46,50,51,52], 22 patients had clear follow-up information (progressive or dead outcomes and duration). Among the 22 patients, 8 patients were not resistant to TKIs before radiotherapy, 6 patients were resistant to 1 line of TKIs (all were imatinib-resistant), 7 patients were resistant to 2 lines of TKIs (all were imatinib- and sunitinib-resistant) and 1 patient was resistant to 3 lines. Since the role of radiotherapy in GISTs should be discussed after excluding the influence of effective TKIs, we analyzed the cases in which radiotherapy was used alone, as well as those in which radiotherapy was used with previously resistant TKIs. There were six patients treated by radiotherapy without any continued TKIs and eight patients treated by radiotherapy with previously resistant TKIs (Table 5). For the six patients with advanced or metastatic GISTs in the abdomen (three), brain (two), and bone (one), there were four patients not resistant to TKIs, one patient resistant to imatinib and sunitinib, and one patient resistant to three lines of TKIs. The median follow-up was 11.5 months (range, 3–72), and there were three deaths (one of the three deaths had a definite progression of irradiated lesions during follow-up). The estimated median PFS was 9 months (Figure 2A). Regarding the eight patients with advanced or metastatic GISTs in the bone (five), brain (one), liver (one), and pararenal and supraclavicular regions (one) treated by radiotherapy with previously resistant TKIs, four patients were imatinib-resistant, and four patients were imatinib- and sunitinib-resistant. There were five deaths, with a median follow-up of 4.5 months (range, 1.5–19). The estimated PFS was 5 months (Figure 2B).

Regarding the 17 undefined irradiated lesions in 17 patients, 13 patients were from 13 case reports [16,18,19,24,25,27,33,36,38,39,40,42,44], and the other 4 patients were from one case series [45]. The four patients with rectal GISTs treated by adjuvant radiotherapy without TKIs after surgery were all alive and had no recurrence during a follow-up of 21–75 months. The remaining 13 lesions in 13 patients included 1 lesion in the rectum, 1 in the stomach, 3 in the bone (1 in the skull and the other 2 in the spine), and 8 in the brain. Among the 13 patients, 8 patients were not resistant to TKIs before radiotherapy, 3 were resistant to 1 line of TKIs (all were imatinib-resistant), 1 was resistant to 2 lines of TKIs (imatinib- and sunitinib-resistant), and 1 was resistant to 4 lines of TKIs. There were 9 patients treated by radiotherapy without continued TKIs after surgery in these 13 patients (Table 6). Two of the nine patients received radiotherapy for the primary lesion areas (rectum and stomach), while the other seven patients received radiotherapy for the brain (five) and spine (two) metastases. Given that metastatic GISTs had a profound impact on prognosis, we performed survival analysis on the seven patients. Among the seven patients, there were four patients not resistant to TKIs, one patient resistant to imatinib, one patient resistant to imatinib and sunitinib, and one patient resistant to imatinib, sunitinib, regorafenib, and dasatinib. There were three deaths (one of the three deaths had definite recurrence during follow-up), with a median follow-up of 6 months (range, 2–24). The estimated median RFS was 20 months (Figure 2C), and the estimated OS was 20 months (Figure 2D).

In addition, there were three other case series, which did not clearly describe the specific scenarios of radiotherapy combined with TKIs [47,48,49]. Joensuu et al. reported 25 patients with advanced or metastatic GISTs receiving radiotherapy, which was a prospective clinical study [47]. PR was seen in 2 patients, and SD was seen in 20. In total, 20 patients progressed, and 18 died, with a median follow-up of 9 months (range, 2–74). The estimated median time to target lesion progression was 16 months, and the median OS was 19 months.

Both interstitial brachytherapy (iBT) and radioembolization are internal irradiation. Omari et al. reported that among 10 imatinib-resistant metastatic GISTs treated with iBT (TKIs continued in 7 patients), 1 recurred, and 6 died during follow-up (range, 2.3–92.9 months), with a median PFS of 6.8 months (range, 3.0–20.2) and a median OS of 37.3 months (range, 11.4–89.7); local tumor control was 97.5% [48]. Rathmann et al. reported nine patients who received radioembolization for liver metastases [49]. CR was seen in three patients, PR in five patients, and SD in one. Eight patients progressed at the end of the study, with a median progression time of 15.9 months (range, 4–29). There were four deaths, and the median OS was 29.8 months (range, 10–72).

### 3.3. Symptom Palliation

We considered symptom palliation in defined lesions treated by radiotherapy alone or radiotherapy with previously resistant TKIs. Among the 41 patients who had defined lesions, 30 patients received radiotherapy alone or radiotherapy with previously resistant TKIs. There were 28 patients with symptomatic GISTs. Local pain, which occurred in 17 patients, was the most common symptom, followed by neurological dysfunction (4) and bleeding (3). There were 22 patients achieving partial or complete palliation. Two patients did not achieve palliation, and the other four patients were not available. The symptom palliation rate was 78.6% (22/28).In addition, the other case series reported symptom palliation in 12 patients with advanced or metastatic GISTs treated by radiotherapy with concomitant TKIs [53]. Pain, spinal cord compression, and bleeding were the main symptoms. There were nine patients who had at least partial palliation.

### 3.4. Adverse Events

There were five case reports and four case series reporting adverse events [11,20,22,25,47,48,49,52,54]. Adverse reactions were reported in 14 patients from 5 case reports and 1 case series with nausea in 3 patients (grade 1), diarrhea in 3 (grade 1–3), fatigue in 3 (grade 1–2), esophagitis in 2 (grade 2), proctitis in 1 (grade 2), chest pain in 1 (grade 1), urinary urgency in 1 (grade 1), dysgeusia in 1 (grade 1), mucositis in 1 (grade 2), dermatitis in 2 (grade 1), and moist desquamation in 1 (grade 2) [11,20,22,25,52,54]. Al-Jarani et al. first reported pericardial cutaneous fistula after radiotherapy in a metastatic GIST patient [54]. In addition, Rathmann et al. reported that among nine GIST patients who received radioembolization for liver metastasis, laboratory findings increased from grade 0 to grade 1 toxicity in seven cases, and stomach ulceration was grade E in one patient according to the Society of Interventional Radiology (SIR) guidelines [49]. Joensuu et al. reported that transient diarrhea was the most common adverse event (52%), followed by pain (44%), nausea (36%), and fatigue (32%) in 25 patients. The adverse events were mainly mild to moderate (grade 1 or 2), and only a few were severe (grade 3). Only one patient developed grade 4 biliary tract necrosis [47]. Omari et al. reported that of 10 patients with imatinib-resistant metastatic GISTs who received iBT, 3 had elevated inflammatory parameters (grade 1), and 2 had local hepatic hemorrhage and pneumothorax (grade 3) [48].

## 4. Discussion

In the era of TKIs, the management of GISTs has undergone revolutionary changes [55,56]. The effectiveness and safety of TKIs have been demonstrated in basic and clinical studies and benefit most GIST patients [57,58,59]. However, we still have to address the problems of secondary resistant mutations. The accurately molecular analysis is the gold standard of GIST diagnosis and also helps patients choose the optimal treatment [60]. wild-type *KIT*/*PDGFRA* and some special mutation sites in GISTs such as *PDGFRA D842 V* result in a limited response to imatinib [61,62,63]. According to the guidelines, the standard treatment for multiple systemic metastases is TKIs [2]. However, there are a considerable number of patients with advanced or metastatic GISTs who need further effective treatment. Therefore, multimodal management of GIST patients (including surgery, radiotherapy, radiofrequency ablation, etc.) has been examined in advanced GISTs [22,64,65,66]. In the past, GISTs were thought to be resistant to radiation therapy [10]. However, some studies have shown that radiotherapy may have some effect on selected GIST patients. Therefore, the effectiveness and role of radiotherapy should be reevaluated.

According to the current systematic review, the quality of most case reports and case series included was low. High-quality research on radiotherapy in GISTs is needed in the future. Compared with many other malignant tumors, GISTs have a better prognosis, and patients are expected to live longer. In many published clinical studies, outcome events often require an extended follow-up [67,68], but most of the patients included in this study had advanced GISTs. Therefore, it is possible to observe the outcome events in a relatively short follow-up period.

We analyzed the defined irradiated lesions, which are presented in Table 3. We divided the lesions into four groups. Two of the four groups were “R” and “R + rT”, in which some lesions achieved objective response when treated by radiotherapy. This may indicate that radiotherapy had some effect on selected patients. The previous view that GISTs were insensitive to radiotherapy may need to be reevaluated.

The latest NCCN guidelines recommend radiotherapy for the palliative treatment of bone metastases [69]. We further analyzed the sites of these defined irradiated lesions and found that bone was the most common site, followed by the abdomen. There were some lesions in the abdomen that achieved objective response and disease stabilization. This indicated that, in addition to bone, abdominal lesions may be treated by radiotherapy, especially if the tumor is relatively fixed in the abdominal cavity [51]. Even in the past, there were concerns about the adverse effects of radiation on the abdominal organs. Furthermore, radiotherapy combined with imatinib should be considered, especially for GISTs at high risk of local recurrence, where surgery is often demolitive, such as rectal and esophageal GISTs [11,22].

According to our study, there were six patients (four patients not resistant to TKIs) with advanced or metastatic GISTs treated by radiotherapy alone. PR was seen in three patients, with an estimated median PFS of 9 months in the six patients. Compared with patients with advanced GISTs treated initially with imatinib [70], radiotherapy may not benefit the survival of patients with advanced GISTs. In addition, eight patients (four patients were imatinib-resistant and four were imatinib- and sunitinib-resistant) received radiotherapy with previously resistant TKIs. CR was seen in one patient, and SD was seen in one patient, with an estimated median PFS of 5 months in the eight patients. Compared with patients treated by further lines of TKIs after imatinib or imatinib and sunitinib failure [71,72], radiotherapy with continuous use of previously resistant TKIs may not benefit survival for patients with advanced GISTs. Thus, TKIs are still the mainstay for advanced or metastatic GISTs. However, radiotherapy may help achieve objective response in selected patients. Therefore, when TKIs are not available, radiotherapy may be an option for some patients. Nevertheless, it may not benefit survival in patients with systematic metastases. In addition, the survival of the “R + rT” group was inferior to that of the “R” group, which may be because all eight patients were drug-resistant. Joensuu et al. reported that 25 patients who were progressive during or after TKIs were treated by radiotherapy. There were 19 patients treated with concomitant TKIs. The study did not further analyze the efficacy of radiotherapy alone or radiotherapy with concomitant previously resistant TKIs. Therefore, the results of the study failed to demonstrate the effectiveness of radiotherapy in GISTs.

There were seven patients with undefined irradiated lesions mainly located in the brain treated by radiotherapy alone. The estimated median RFS was 20 months. Sym et al. reported that compared with imatinib-resistant patients after surgery, patients responsive to imatinib had better survival after surgery [73]. The results should be interpreted with caution. Among the seven patients in our study, four patients were not resistant to TKIs and may benefit from the surgery. In addition, some patients might have been contaminated by TKIs after surgery, which was not reported. Meanwhile, previous studies have indicated that surgery should be chosen with greater caution in patients with multiple systemic metastases [48]. In cases of limited disease progression after TKIs, a more aggressive approach can be chosen [74,75], but the risks of surgical complications and potential benefits cannot be quantified [76,77]. Some studies have also pointed out that complete surgical resection has a significant impact on the survival of GIST patients. In contrast, adjuvant radiotherapy has no apparent benefit except for controlling the target area [78,79]. For locally advanced GISTs, previous studies have evaluated the effectiveness of TKIs in neoadjuvant settings [80]. However, there have been a few case reports about neoadjuvant radiotherapy in GISTs, which may need further investigation.

In addition, relatively high disease control has been achieved in radioembolization and iBT [48,49], which suggests that the two special radiation means may be superior to others for local tumor control. However, hepatopulmonary shunt and radiation pneumonia may limit the use of radioembolization [81].

Through the analysis of symptomatic GISTs, we found that the symptom palliation rate of radiotherapy alone and radiotherapy with concomitant previously resistant TKIs reached 78.6% (22/28), which supports the application of radiotherapy in GISTs for palliative purposes recommended by the guidelines [2]. In addition to pain, radiotherapy may also be used to relieve the symptoms of bleeding and spinal cord compression [51,53]. Patterson et al. reported that among 12 patients with advanced or metastatic GISTs treated by radiotherapy, 9 had improvement in symptoms to varying degrees [53]. However, all 12 patients received TKIs during radiotherapy. It was not clear whether further lines of TKIs after resistance or sensitive or resistant TKIs were used. Thus, the role of radiotherapy in symptom palliation was not clearly explained.

Radiotherapy with concomitant TKIs was well-tolerated. Most adverse events were grade 1–2. However, some adverse reactions suggested that we need to be cautious in simultaneous treatment. TKIs that inhibit *VEGF* receptors may be associated with local dermal toxicity and hepatotoxicity at irradiated sites [82,83,84,85].

Regarding the modes of radiotherapy, Joensuu et al. reported radiotherapy for liver and abdominal tumors, for which three-dimensional (3D) conformational radiotherapy and intensity-modulated radiotherapy (IMRT) were mainly used [47]. These methods belong to the category of stereotactic radiotherapy. Stereotactic radiotherapy has the advantages of precise localization, the concentration of dose, minor impact on the surrounding tissue of tumors, and a high ablative dose. Radioembolization and iBT have also shown good efficacy and safety in treating liver metastases [48,49]. These radiotherapy methods can deliver a relatively high dose to target lesions and can protect the important surrounding structures, ultimately achieving a better response [11,86,87,88].

## 5. Limitations

We must acknowledge that this study has several limitations.

Most importantly, publication bias was present in the current study. Because GISTs were considered insensitive to radiotherapy in the past, positive results of radiotherapy treatment had a tendency to be published, which may overstate our findings. Furthermore, with the wide application of TKIs, radiotherapy was rarely considered in GISTs, resulting in few publications reporting radiotherapy in GISTs. However, to discuss the application of radiotherapy in GISTs more comprehensively, we conducted an extensive literature search and included almost all the positive and negative cases of radiotherapy that could be retrieved. A considerable number of patients were not responsive to radiotherapy in our study. However, there were still many negative cases that could not be obtained through the literature search. Further research is, therefore, necessary on radiotherapy in GISTs.

Second, the articles included were low- to moderate-quality case reports and case series. In addition, there was data heterogeneity in these studies. Thus, in the future, we need to design high-quality randomized controlled studies to evaluate the efficacy and safety of radiotherapy in GISTs.

Third, in the era of TKIs, radiotherapy with concomitant TKIs has been more common, which may mean that the effects of radiotherapy cannot be effectively evaluated. Thus, it is necessary to strictly design research to assess the value of radiotherapy in GISTs from an ethical point of view.

## 6. Conclusions

Overall, radiotherapy may relieve symptoms for some GIST patients with advanced or metastatic lesions and even help achieve objective response in selected patients without significantly reducing the quality of life. In addition to bone metastases, fixed abdominal lesions may be treated by radiotherapy.

Nevertheless, the efficacy and safety of radiotherapy in GIST patients warrant further investigation.

## Figures and Tables

**Figure 1 cancers-14-03169-f001:**
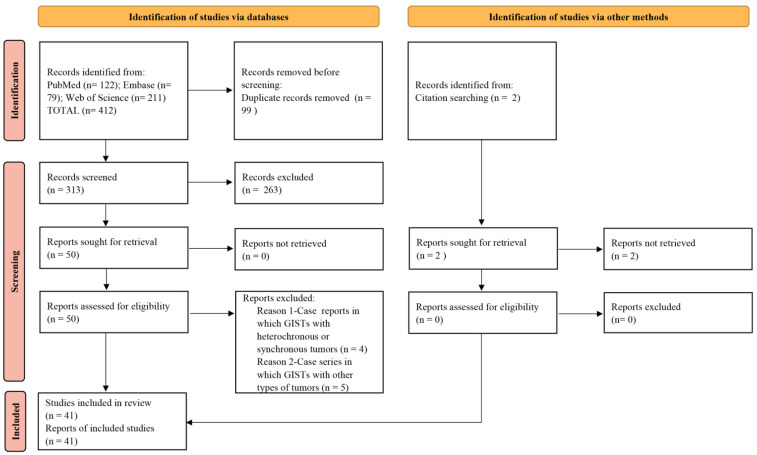
Flow diagram of the literature search and selection criteria.

**Figure 2 cancers-14-03169-f002:**
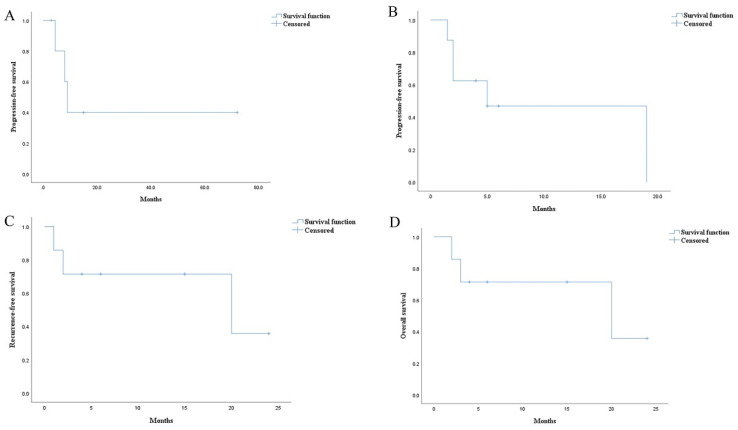
Kaplan–Meier estimates of survival: (**A**) progression-free survival in the 6 patients treated by radiotherapy without any continued TKIs; (**B**) progression-free survival in the 8 patients treated by radiotherapy with previously resistant TKIs; (**C**) recurrence-free survival in the 7 patients treated by radiotherapy without continued TKIs after surgery; (**D**) overall survival in the 7 patients treated by radiotherapy without continued TKIs after surgery.

**Table 1 cancers-14-03169-t001:** Description of the cases treated by radiotherapy.

References	Age/Gender	Location	Previous TKIs	Means/Total Dose*Fractions ^#^	Concomitant TKIs	Palliation	Response	Side Effects	Follow-Up (Recurrence or Progression/TTR or TTP (mo)/A or D/OS (mo)
Shioyama et al., 2001 [50]	75/female	retroperitoneum	None	R + C + I/51Gy*34	None	Yes	PR	NA	No/72/A/72
Pollock et al., 2001 [25]	77/female	rectum	None	S + R/50.4Gy*NA	None	Yes	-	desquamation of perineum/grade 2	No/18/A/18
Akiyama et al., 2004 [37]	60/male	around the left optic nerve	None	R/54Gy*18	None	Yes	NA	NA	NA/NA/D/4.5
Puri et al., 2006 [44]	42/male	right parietal lobe	None	S + R + C/60Gy*NA	None	Yes	-	NA	No/20/D/20
Boruban et al., 2007 [23]	55/male	pelvic	None	S(I) + R + T/54Gy*27	imatinib	Yes	CR	NA	No/37/A/37
Barrière et al., 2009 [41]	57/male	clivus/lumbar spine	imatinib/sunitinib (R)	R + T/NA	sunitinib/nilotinib	No	NA	NA	NA/NA/D/5
Ciresa et al., 2009 [52]	54/male	rectum	None	R + T/37.8Gy*21	imatinib	Yes	PR	neutropenia/grade 3/proctitis/grade 2	NA/NA/NA/NA
Hamada et al., 2010 [27]	54/female	left frontal lobe	imatinib(W)	S + R/NA	None	NA	-	NA	No/6/A/6
Tezcan et al., 2011 [30]	83/male	right femur head	None	R + T/30Gy*10	imatinib	Yes	NA	NA	NA/NA/A/NA
Knowlton et al., 2011 [16]	37/male	stomach	None	S + R/36Gy*24	None	Yes	-	No	No/240/D/240
Naoe et al., 2011 [19]	77/female	right cerebral peduncle/left occipital lobe	None	R + T/NA/S + R + T/NA	Imatinib (I)	NA	NA/-	NA	No/2/D/2
Lolli et al., 2011 [10]	48/female	left supraclavicular	imatinib/sunitinib/nilotinib/sorafenib (R)	R + T/50Gy*25	sorafenib	Yes	SD	well tolerated	No/NA/A/NA
Di Scioscio et al., 2011 [29]	62/male	spine	None	R + T/30Gy*NA	imatinib	Yes	NA	NA	Yes/24/D/34
Abuzakhm et al., 2011 [34]	57/female	left humerus	imatinib/sunitinib (R)	R + T/NA	sunitinib	NA	NA	NA	NA/NA/D/2
Wong et al., 2011 [38]	26/male	left frontal temporal	imatinib/sunitinib (R)	S + R/NA	None	NA	-	NA	No/4/A/4
Slimack et al., 2012 [36]	37/male	spine	imatinib (R)	S + R + C/NA	None	Yes	-	NA	No/24/A//24
Halpern et al., 2012 [28]	62/male	right upper quadrant/retroperitoneum	imatinib (I)	R/63.4Gy*NA	None	Yes	PR	well tolerated	No/3/A/3
Feki et al., 2012 [35]	58/male	sternoclavicular joint	None	R + T/30Gy*NA	imatinib	Yes	PR	NA	No/10/A/19
Drazin et al., 2013 [39]	60/male	left frontal lobe/left cerebellum	None	R/18Gy*1/S + R/NA	None	Yes	NA/-	NA	No/15/A/15
Takeuchi et al., 2014 [26]	74/male	right lateral ventricle	imatinib/sunitinib (R)	R + T/NA	sunitinib	-	CR	NA	No/4/A/4
Sato et al., 2014 [40]	80/male	vermis	None	S + R/22Gy*11	None	Yes	-	NA	Yes/1/D/3
Aktan et al., 2015 [31]	56/male	right femur/L1–3 vertebrae	Imatinib (R)	R + T/30Gy*10	imatinib	Yes	NA	NA	NA/NA/D/2
	70/male	L2 vertebra	Imatinib (R)	R + T/30Gy*NA	imatinib	Yes	NA	NA	NA/NA/D/1.5
Gupta et al., 2016 [24]	64/female	right frontal skull	Imatinib (R)	S + R + T/35Gy*14	imatinib/sunitinib	Yes	-	NA	Yes/21/A/24
Gatto et al., 2017 [22]	62/male	paracaval lesion	imatinib/sunitinib (R)	R + T/35Gy*14	regorafenib	Yes	PR	No	No/36/A/36
	44/male	pararenal/supraclavicular	imatinib/sunitinib (R)	R/85Gy*9/R + T/32Gy*5	sunitinib	Yes	SD	nausea/NA	No/5/A/5
Loaiza-Bonilla et al., 2017 [46]	35/male	liver/right retropharyngeal	imatinib (R)	R + T/NA	regorafenib	-	SD	NA	NA/NA/A/3
Badri et al., 2018 [42]	66/male	right cerebellum	None	S + R/NA	NA	NA	-	NA	No/12/A/12
Jang et al., 2018 [43]	70/male	liver	Imatinib (R)	E + R + T/40Gy*16	Imatinib	Yes	NA	NA	No/6/A/6
Yang et al., 2018 [51]	74/male	duodenal bulb	imatinib/sunitinib (R)	R/32.5Gy*13	None	Yes	PR	NA	Yes/9/D/16
Katayanagi et al., 2019 [32]	56/male	T8 vertebra/right ilium	imatinib/sunitinib (R)	R + T/37.5Gy*15	sunitinib/imatinib	NA	NA	NA	NA/NA/D/19
Yilmaz et al., 2020 [17]	31/male	right iliac bone	imatinib (R)	R + T/24Gy*3	sunitinib	Yes	CR	No	No/16/A/16
Carvalho et al., 2020 [18]	76/female	left frontal lobe/right cerebellar	imatinib (R)	S + R + T/NA/R + T/NA	imatinib	No	-/NA	NA	NA/NA/D/6
Andruska et al., 2020 [21]	29/female	caudate lobe of liver	imatinib/sunitinib/sorafenib/regorafenib (R)	R + T/30Gy*10	regorafenib/sunitinib	-	NA	NA	NA/NA/D/NA
Lo et al., 2020 [33]	63/male	T9 vertebra	imatinib/sunitinib/regorafenib/dasatinib (R)	S + R/30Gy*10	None	NA	-	NA	NA/NA/D/2
Maria et al., 2022 [20]	77/male	left maxillary	imatinib/2 additional lines (R)	R/35Gy*10	None	Yes	PR	mucositis/grade2/dermatitis/grade1	NA/NA/D/8
Al-Jarani et al., 2022 [54]	52/female	liver/xiphoid	NA	NA	NA	-	NA	change in skin, dermatitis, sclerosis, fistula/NA	NA/NA/A/48

^#^ Total dose and fractions; Abbreviations: T, tyrosine kinase inhibitors (TKIs); S, surgery; R, radiotherapy; C, chemotherapy; E, embolization; I, immunotherapy; NA, not available; CR, complete response; PR, partial response; SD, stable disease; PD, progressive disease; (R), resistance; (I), intolerance; (W), withdrawal; Gy, gray; TTP, time to progression; TTR, time to recurrence; (mo), month; A, alive; D, dead.

**Table 2 cancers-14-03169-t002:** Description of the case series treated by radiotherapy.

References	Sex, Total No. (Male/Female)	Age, Median (Range),y	Sites	Previous TKIs, Patients No.	Means/ Dose Range	Concomitant TKIs, Patients No.	Symptom Palliation, Patients No.	Response	Follow-Up, Range (mo)/Outcome
Baik et al., 2007 [45]	4 (1/3)	53 (41–68)	Rectum	None	R/45–54 Gy	None	NA	-	21–75/No recurrence and all alive
Cuaron et al., 2013 [11]	15 (8/7)	68 (41–86)	Bone/Abdomen/Pelvis,	11	R/15–50 Gy	5	12	PR in 5 patients, SD in 9	1.4–28.3/12 deaths
Joensuu et al., 2015 [47]	25 (17/8)	61.4 (19.7–86.5)	Abdomen	25	R/30–40 Gy	19	NA	PR in 2 patients, SD in 20	2–74/20 patients progressed and 18 deaths
Rathmann et al., 2015 [49]	9 (7/2)	55 (34–74)	Liver	9	RE/0.55–1.88 Gbq	9	NA	CR in 3 patients, PR in 5, SD in 1	10–72/8 progressed and 4 deaths
Omari et al., 2019 [48]	10 (9/1)	58.5 (37–68)	Liver/Peritoneum	10	iBT/6.7–22.0 Gy	7	NA	LTC 97.5%	2.3–92.9/one relapse and 6 deaths
Patterson et al., 2022 [53]	12 (7/5)	69 (36–79)	NA	NA	R/20-50 Gy	12	9	SD in 1 patient, PD in 1	NA

Abbreviations: T, tyrosine kinase inhibitors (TKIs); mo, month; S, surgery; R, radiotherapy; RE, radioembolization; iBT, interstitial brachytherapy; Gy, gray; NA, not available; LTC, local tumor control; CR, complete response; PR, partial response; SD, stable disease.

**Table 3 cancers-14-03169-t003:** Response to radiotherapy with/without concomitant TKIs in the definite irradiated lesions.

Response	R	R + nT	R + rT	R + sT
CR	0	1	1	1
PR	6	1	4	2
SD	6	3	5	0
PD	2	0	0	0
NA	5	1	12	3
N	19	6	22	6

Abbreviations: CR, complete response; PR, partial response; SD, stable disease; PD, progressive disease; NA, not available; N, number; R, radiotherapy; R + nT, radiotherapy with new TKIs (radiotherapy with further lines of TKIs after resistance); R + rT, radiotherapy with resistant TKIs (radiotherapy with previously resistant TKIs); R + sT, radiotherapy with sensitive TKIs (radiotherapy with imatinib in cases in which no TKIs have been used before).

**Table 4 cancers-14-03169-t004:** Response of GIST at different locations to radiotherapy.

Response	Brain	Neck	Chest	Abdomen	Pelvis	Bone and Joint	N
CR	1				1	1	3
PR			1	6	1	5	13
SD		3	1	5	1	4	14
PD						2	2
NA	4			3		14	21
N	5	3	2	14	3	26	53

Abbreviations: GISTs, gastrointestinal stromal tumors; CR, complete response; PR, partial response; SD, stable disease; PD, progressive disease; N, number.

**Table 5 cancers-14-03169-t005:** Application of TKIs in 22 patients with defined lesions before and after radiotherapy.

Continued TKIs	Resistant to 0 TKIs	Resistant to 1 TKI	Resistant to 2 TKIs	Resistant to ≥3TKIs
None	4		1	1
rTKI	-	4	4	
nTKI	-	2	2	
sTKI	4	-	-	-
NA				
N	8	6	7	1

Abbreviations: TKIs: tyrosine kinase Inhibitors; rTKI, previously resistant TKIs; nTKI, further lines of TKIs after resistance; sTKI, imatinib in cases in which no TKIs were previously used; NA, not available; N, number.

**Table 6 cancers-14-03169-t006:** Application of TKIs in 13 patients with undefined lesions before and after radiotherapy.

Continued TKIs	Resistant to 0 TKIs	Resistant to 1 TKI	Resistant to 2 TKIs	Resistant to ≥3 TKIs
None	6	1	1	1
rTKI	-	2		
nTKI	-			
sTKI	1	-	-	-
NA	1			
N	8	3	1	1

Abbreviations: TKIs: tyrosine kinase Inhibitors; rTKI, previously resistant TKIs; nTKI, further lines of TKIs after resistance; sTKI, imatinib in cases in which no TKIs were previously used; NA, not available; N, number.

## Data Availability

The paper has already contained all date sets.

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
