# Peer review of "Radiotherapy in the Management of Gastrointestinal Stromal Tumors: A Systematic Review"

_cancers, 2022, doi:10.3390/cancers14133169_

Round 1
Reviewer 1 Report
This manuscript has been largely revised so it is good for publication. However, some point should be addressed before publication.
1. Table S2 is not cited in the main text and this table is not completed.
2. There were 35 case reports and 6 case series in section 3.1 is compatible with Figure 1. But there were only 34 case reports and 2 case series in section 3.2.
3. This study repeatedly mentioned about the RTO in GIST patients after exhausting available TKIs. However, 22 patients from case reports with clear follow-up information and some of them did not meet the criteria of exhausting available TKIs.
4. Some grammar mistakes should be checked and corrected. For example, present à presented (Line 200)
5. I suggested added some comments and discussion of some points to make this article more comprehensive.
May consider cite the referece in terms of molecular diagnosis of GIST. (PMID: 31100836)
May discuss the role of local surgery in addition to RTO (PMID: 33976743, PMID: 28448384)
May discus the role of RTO in neoadjuvant setting in addition to TKIs (PMID: 30934606)
Reviewer 2 Report
Zhang et al presented the title, "Radiotherapy in the management of gastrointestinal stromal tumors: A systematic review" which is a systematic study, is a well-written manuscript in the interest of readers.
Although the paper has enough elements to consider in the current journal if the authors revise the minor changes mentioned below.
There are typographical errors in the manuscript, please make sure to have a thorough check.
say, e.g Page 2, line 54, "GISTs[6,7]", there should be a gap between the end word of a sentence and a full stop sign.
Please follow the guidelines provided by MDPI Cancer Journal.
Author Response
Response: We have carefully checked the full manuscript and revised typographical errors in it according to guidelines provided by MDPI Cancer Journal.
Reviewer 3 Report
Gastrointestinal stromal tumors are important clinical problem. Most of them are insensitive to radiotherapy. Theaim of this study was to pay attention of scholars and clinicians to radiotherapy in gastrointestinal stromal tumors.
Author Response
Response: Thank you very much for your comments.
This manuscript is a resubmission of an earlier submission. The following is a list of the peer review reports and author responses from that submission.
Round 1
Reviewer 1 Report
This manuscript entitled “Radiotherapy in the management of gastrointestinal stromal tumours: A systematic review” presented a systematic review of efficacy of toxicity for GIST patients treated with RT. They did a very comprehensive literature review, but the major limitation exists.
As the authors mentioned that GIST is RT-insensitive so majority of GIST patients can not benefit from RT. However, very small population of cases experiencing good response were presented as case reports. I do not consider that the cases without response to RT can be published!! In this review, most cases were from case reports, therefore, the selection bias largely exist in the review and the efficacy of RT has been largely overestimated.
Reviewer 2 Report
This manuscript is a systematic review article that summarized and analyzed the previous reports which described treatments using radiotherapy in patients with GIST. The authors showed that the disease control was 91-93%, the objective response was 29-36%, and the estimated median progression-free survival was 19 months with few severe toxicities. Furthermore, the authors demonstrated that radiotherapy was useful for symptom palliation. This topic is interesting and informative to researchers and clinicians in the field as usefulness of radiotherapy for GIST has not been fully investigated yet.
However, the following major and minor issues require clarification:
Major
- The extracted reports are too heterogeneous to make the assessment of radiotherapy. Especially, as significant proportion of cases which prescribed concomitant tyrosine kinase inhibitors is included in the analysis, it’s questionable if radiotherapy is sensitive and effective to GIST. I recommend that the authors analyze the efficacy of radiotherapy with excluding the cases using concomitant TKIs.
- Table 1 is too large and contains too much information. It should be more summarized as readers can easily refer to it.
Minor
- Planning of radiotherapy should be described, including dose fractions, biological effective doses and the use of three-dimensional conformal RT.
- Some of “gastrointestinal stromal tumor” are not abbreviated.
- (Discussion) The data heterogeneity and few high-quality studies should be included in the limitations.